# Discrete Predictive Models for Stability Analysis of Power Supply Systems

**Natalia Bakhtadze \*** and **Igor Yadikin**

V.A. Trapeznikov Institute of Control Sciences, 65, Profsoyuznaya, 117997 Moscow, Russia; jadikin1@mail.ru
\*   Correspondence: sung7@yandex.ru; Tel.: +7-916-544-2259

**Abstract:** The paper offers an approach to the investigation of the dynamics of nonlinear non-stationary processes with the focus on the risk of dynamic system stability loss. The risk is assessed on the basis of the accumulated knowledge about power supply system operation. New methods for power supply modes analysis are developed and applied as follows: linear discrete point knowledge-based models are developed for nonlinear non-stationary objects; wavelet analysis is used for non-stationary processes; stability loss risks are analyzed through the investigation of spectral decompositions of Gramians of these linear predictive models. Case studies are included.

**Keywords:** process identification; knowledgebase; associative search models; wavelet analysis; Gramian method

---

## 1. Introduction

In order to gain competitive advantages, industrial enterprises have to modernize their equipment and introduce modern information technologies. A reliable power supply with consistently high power quality is key to higher productivity, especially where AC motors are the key consumers. The ability to predict process approach to stability thresholds is particularly important.

### 1.1. Problem Background

This study aims at the development of a systematic approach to assessing the risks of dynamic systems stability loss based on the use of accumulated knowledge about the operation of power supply systems. It also focuses on new methods for analyzing power supply modes planning. New methods comprise linear point knowledge-based predictive models for a wide class of nonlinear non-stationary objects, wavelet analysis and spectral decompositions of Gramians for those linear models.

Matrix Lyapunov equations play an important role in modern mathematics, including the mathematical control theory [1,2]. The properties of their solutions are closely related with the structural properties of controllability and observability that should be taken into account when analyzing the static and transient stability of power systems [3–6]. Spectral methods have been widely used over the past decades for studying the stability of linear and, to a lesser extent, nonlinear discrete and continuous systems [7–11]. They are used for reducing the dimension of mathematical models of power systems (model order reduction, MOR) [8,10,11] and for optimal global control of electric power systems [3].

Further approaches to solving MOR problems are based on minimizing the square of $H_2$-norm of the residual and use the solution of the matrix Lyapunov equations [9]. Static and transient stability in power systems are typically investigated with the help of linear and selective linear modal analysis [5,11], the method of normal forms NF [3], the method of modal series [6] and the Koopman Mode Decomposition (KMD) method.

*1.2. Main Objectives and Research Methodology*

The methods listed in the precious subsection are typically used with linear models that is not always acceptable for many applications. Moreover, the available a priori information about the control object may be insufficient for achieving the required control performance with the help of traditional methods. In such cases, the application of intelligent modeling techniques based on e-learning is preferable, especially for nonlinear, poorly structured or poorly formalized systems. The development and application of models using knowledge bases for automatic control systems are becoming increasingly relevant.

The paper shows the possibility of constructing high-precision linear point (i.e., virtual in the "here and now" sense) models of nonlinear objects capable of describing processes dynamics in electric power systems. These linear models created anew at each time step use all available information about the plant both a priori and statistical. This enables the development of high-precision models even for essentially nonlinear and time-varying systems.

Knowledge can be interpreted as patterns extracted from process data through data mining. In the approach proposed by the authors, knowledge is formalized in the form of templates of input and output variables that describe process operation. Thus, our method for analyzing the stability of nonlinear systems can be briefly formulated as follows.

1. At the first stage, a point (the adjective "point" means here "for a certain time step") linear model of a nonlinear process is created. By data mining, the values of the input variables (in the general case, a vector) are selected from the process history that are close to the current value of the input variable subject to a certain criterion. The selected values of the historical data correspond to the actual values of the outputs [12–14]. Further, a consistent system of linear equations is formed. The only solution to this system by the least squares method gives the values of the linear model coefficients as well as the forecast of the output at the next time step. The described identification method for a nonlinear dynamic object may be called "intelligent least squares".

2. At the second stage, the state of the system described by the point linear predictive model is investigated. The paper discusses in detail the capabilities of the Gramian method for this purpose. The main criterion of the system's state approach to stability threshold is the unlimited growth of the Frobenius norm for controllability Gramian. To that end, the energy of weakly stable modes is investigated. It is calculated based on the spectral decomposition of the Gramians or the spectral decomposition of the quadratic Frobenius norm of the system's transfer function. We study the weakly stable eigenvalues of the dynamics matrix of a linear (point) model of a dynamic system [15–18]. Comparing this energy with the system's total momentum energy makes it possible to predict the potential center of unstable oscillation for the trajectories resulting in the cascade failure. It was ascertained in [16,18] that for this purpose it is enough to investigate the Frobenius norm of the transfer function.

Another method for studying the stability of a nonlinear system is the analysis of the wavelet spectrum of its linear model constructed with the help of the associative search technique. The paper offers a brief description of this approach and stability criteria. This method is less detailed than the Gramian's method; however, it gives good results for time varying systems.

Further text is organized as follows.

Section 2 focuses on the associative search models, i.e., the "point" identification models based on data mining. In Section 2.1, the problem is stated. In Section 2.2, the approach to intelligent model development based on inductive learning and knowledge formation is discussed.

Associative search procedure is described in Section 2.3, Section 2.4 discusses how the system of linear equations can be solved in close loop with the LS method, in particular, for control systems with identifier. Sections 2.5 and 2.6 outline the clustering-based associative search. Section 2.7 describes the wavelet approach to the associative search technique for time-invariant processes. Section 2.8 investigates the associative model stability conditions in the sense of multi-scale wavelet expansion for spectrum analysis.

Section 3 outlines the approach to the stability analysis of predictive model based on the Gramian method.

Section 4 examines two application cases: automatic remote diagnostics of power system readiness to general primary frequency control (Section 4.1) and full power prediction (Section 4.2). Finally, the Conclusion summarizes the main results and achievements.

## 2. Knowledge-Based Virtual Identification Models Research Methodology

The use of adjustable identification models has been a long-term trend in the identification theory and applications. The convergence of the empirical distribution functional to the theoretical one with the sample size increase has been analyzed [19].

Identification methods based on data mining [20] not only helped to overcome the challenges posed by nonlinear and non-stationary behavior but also increased significantly the accuracy of analysis and forecasting. Associative search algorithms develop point linear models of nonlinear processes based on process history analysis. Thus, the dynamic adjustment of identification models is carried out on the basis of e-learning, knowledge extraction and replenishment.

These intelligent identification algorithms have shown high accuracy results in various applications, such as chemistry, oil refining, smart grids, and transport systems [21]. To analyze and predict the stability of dynamical systems, an approach based on the study of multiscale wavelet expansion can be used [22].

### 2.1. Predictive Intelligent Model Design Problem Statement

In knowledge-based modeling methods, knowledge is used for the most accurate reproduction of the object's image from its fragment [23]. In the associative search method, knowledge is understood as an associative relationship between images. Set patterns of input and output variables are considered as images. In [22], a model of associative thinking was presented. The memorization process is interpreted as a sequential formation of associations of image pairs.

The model can be considered as an intermediate stage between logical models and neural networks. In our case (associative search), we use pairs of patterns of input and output variables of a dynamic system. At each time step, a new virtual model is created. In order to build a model for a specific time step, a temporary "ad hoc" database of historic and current process data is generated. After calculating the output forecast based on plant's current state, this database is deleted without saving. The linear dynamic prediction model looks as follows:

$$y_t = a_0 + \sum_{i=1}^{r} a_i y_{t-i} + \sum_{j=1}^{s} \sum_{p=1}^{P} b_{j,k} x_{t-j,p} \ \ \forall j = \overline{1, S}, \tag{1}$$

where $x_t$ is the input vector value; $y_t$ is the predicted output value for the next time step; $r$ and $s$ are the output and input memory depth, accordingly; $P$ is the input vector length.

The identification algorithm forms an approximating hypersurface of the input vector space and one-dimensional outputs of the dynamic object (Figure 1). The values of the coefficients of the linear model and the predicted output value are determined on the basis of the least squares method.

In fact, the associative search method simulates the decision-making process of a human operator. The sum

$$d_{t,t-j} = \sum_{p=1}^{P} \lfloor x_{t,p} - x_{t-j,p} \rfloor \ \ \forall j = \overline{1, S} \tag{2}$$

can be considered as a metric in the $P$-dimensional input space, where $j$ is typically less than $t$; $x_{t,p}$ are the components of the input vector at the time step $t$. Source: authors' illustration.

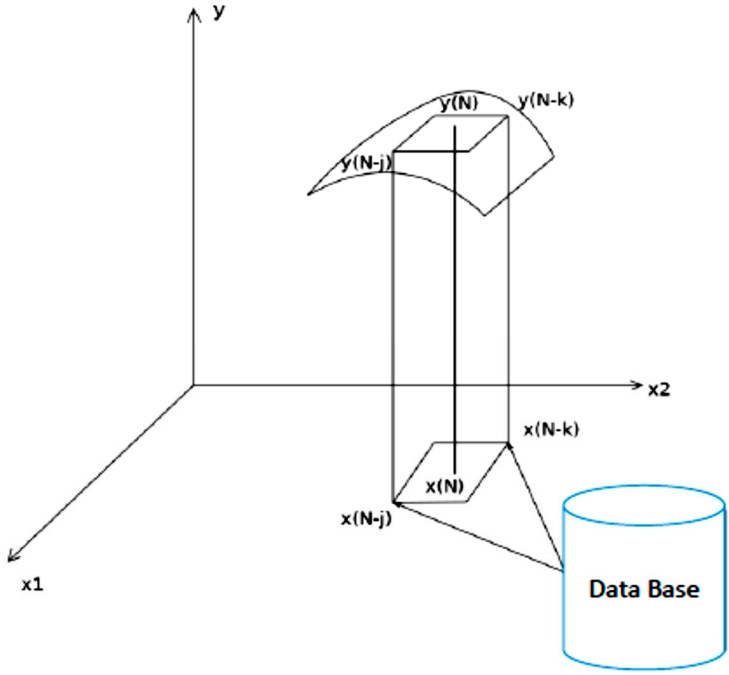

**Figure 1.** Approximating hypersurface design.

*2.2. Development of a Virtual Model on the Basis of Associative Search Technique*

Assume that for the current input vector $x_t$:

$$\sum_{p=1}^{P}\left|x_{t,p}\right| = d_t. \tag{3}$$

In order to build an approximating hypersurface for $x_t$, we select such vectors $x_{t-j}, j = 1, \ldots, s$ from the input data archive that for a given $D_t$ the following conditions will hold:

$$d_{t,t-j} \le d_t + \sum_{p=1}^{P}\left|x_{t-j,p}\right| \le d_t + D_t, \ \ j = 1, \ldots, s. \tag{4}$$

The 2-D case is presented in Figure 2. Source: authors' illustration.

If the domain $D_j$ selected on the basis of knowledge does not allow constructing a simultaneous system of linear algebraic equations, the inputs selection criterion is weakened by increasing the threshold $D_j$.

The purpose of associative search is to restore all features of the object based on the accumulated set of its images, the "dynamic twins". Let $R_0$ be the image that initiates the search; $R$ is the resulting image of the associative search. The pair $(R_0, R)$ may be called *association*. The set of all associations on a set of images makes the content of the intelligent system's knowledgebase.

To apply the associative search technique, a preliminary training stage is required, for which an archive of images is created. The algorithm that implements the $R^a$ image reconstruction procedure based on $R_0^a$ can generally be described by the predicate $(R_{0,i}^a, R_i^a, T^a)$, where $R_{0i}^a \subseteq R_0, R_i^a \subseteq R$. In particular, this predicate can be a function that asserts the truth or falsity of the input vector membership in a certain area of the input space.

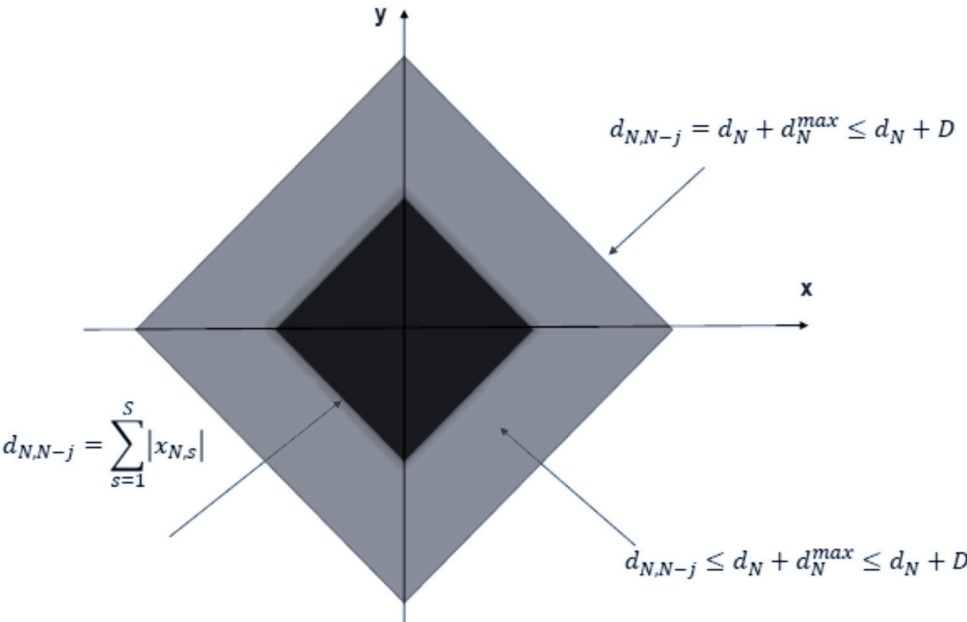

**Figure 2.** Building the approximating hypersurface.

### 2.3. Associative Search Technique in Short-Term Prediction

The associative search technique consists of two sequential stages: (i) the hypersurface is selected in such way that it contains input vector $x_{t-j}, j = 1, \dots, s$ at the current time step $t$, and (ii) the hypersurface is selected from the historical data archive corresponding to $x_{N-j-1}, j = 1, \dots, s$, contains the input vector at the previous time step $t - 1$. The predicate describing the choice will have the form:

$$\Xi\left(R_0^a, R^a, T^a\right) = \left\{ \sum_{p=1}^{P} \left|x_{t-j,p}\right| \le D_t - \sum_{p=1}^{P} \left|x_{t,p}\right|; \quad \sum_{p=1}^{P} \left|x_{t-j-1,p}\right| \le D_{t-1} - \sum_{p=1}^{P} \left|x_{t,p}\right| \right\}. \tag{5}$$

Within the framework of this approach, there is a possibility to improve the accuracy of the procedure by increasing the memory, for example, to $m$ steps ($m < t$):

$$\Xi\left(R_0^a, R^a, T^a\right) = \left\{ \begin{array}{c} \sum_{p=1}^{P} \left|x_{t-j,p}\right| \le D_t - \sum_{p=1}^{P} \left|x_{t,p}\right|; \\ \sum_{p=1}^{P} \left|x_{t-j-1,p}\right| \le D_{t-1} - \sum_{p=1}^{P} \left|x_{t,p}\right|; \dots \\ \sum_{p=1}^{P} \left|x_{t-j-l,p}\right| \le D_{t-l} - \sum_{p=1}^{P} \left|x_{t,p}\right| \end{array} \right\}. \tag{6}$$

### 2.4. Solving the System of Linear Equations for the LS Method

The development of identification models with the help associative search algorithms in the closed loop is, however, not always possible: The control faces the same challenges as the traditional methods do. Dependent values are used in the closed-loop control; optimal controllers generate linear state feedbacks that results in the degenerate problem [24]. In this case, one can apply the Moore–Penrose method and the Singular Value Decomposition (SVD) [25,26] that allows to obtain pseudo-solutions of a system of linear equations in order to apply the associative search method [22].

### 2.5. Associative Search Based on Clustering

The search of data making the best fit to the current values of the input variables may be exhaustive. In order to improve the algorithm performance, one of the clustering methods can be applied. Such methods allow to determine for the current time step the membership of the current input vector in

a certain area of the multidimensional space. Further on, when using the associative search technique, the input vectors close to the current one are selected within a specific cluster.

If the input vector is known to belong to a certain cluster, "close" (similar) vectors are selected within the cluster $C_r^{x_i}$, $i$, and the associative search procedure for $C_r^{x_i}$, $i = 1, \ldots, P$ is applied, where $P$ is the dimension of the input vector $r = 1, \ldots$; $R$ is the number of the cluster in the sub-space $X \in R^p$; $X$ is the set of inputs values:

$$
\begin{aligned}
X = \left\{ C_r^{x_i}, \ i = 1, \ldots, P, \ r = 1, \ldots, R \right\} : \ & \left\{ x_1(t) \in C_r^{x_1}, x_1(t-1) \in C_r^{x_1}, \ldots, x_1(t_0) \in C_r^{x_1} \right\}, \\
& \left\{ x_2(t) \in C_r^{x_2}, x_2(t-1) \in C_r^{x_2}, \ldots, x_2(t_0) \in C_r^{x_2} \right\}, \ldots, \\
& \left\{ x_p(t) \in C_r^{x_p}, x_p(t-1) \in C_r^{x_p}, \ldots, x_p(t_0) \in C_r^{x_p} \right\}.
\end{aligned}
\tag{7}
$$

This approach does not require the knowledge of the nonlinear object structure.

## 2.6. Clusterizaton and Associative Search

Clustering acts as a convenient learning and pre-learning tool that allows to increase the computational performance of associative search.

Crisp and fuzzy approaches may be considered. In the first case, each input vector belongs to only one of the disjoint sets (clusters) of the input space. In case of fuzzy clustering, an object can belong to several clusters simultaneously with various degrees of confidence. The degree of confidence is determined by the selected membership function.

The associative search technique is as follows. The current input vector refers to a certain cluster subject to the criteria of the minimum distance from the center:

$$
\min_k \sum_{k=1}^{K} \| g_k - \bar{x}_N \|^2,
\tag{8}
$$

where $\bar{x}_N \in X$ is the current input vector of the control plant; $g_k$ is the center of the cluster $k$.

For associative search, the vectors close to the current input vector are selected within this cluster. If they are not enough, the cluster can be expanded using single-channel methods that combine two clusters with a minimum distance between members.

## 2.7. Wavelet Analysis of Time-Varying Processes

The recent years have been seeing the growing popularity of time-variant dynamic process analysis based on the wavelet transform. The wavelet transform is a generalization of spectral analysis with respect to Fourier transform. The first works on wavelet analysis examined time series [27]; the method was then considered as an alternative to Fourier transform with frequency localization.

Today, wavelet analysis is extensively used in many areas [28]. The most popular applications include processing and synthesis of non-stationary signals, information compression and coding, image processing, and pattern recognition, particularly in medicine. The method is effective for studying geophysical fields and meteorological time series, as well as for earthquake prediction.

Wavelet analysis is based on a linear transform (called a *wavelet transform*) made by means of soliton-like functions (wavelets) that form an orthonormal basis in $L^2$. These basis functions are localized in a limited area. Therefore, the wavelet transform allows, as against Fourier transform, to obtain information on local properties of the signal. Wavelets also provide a powerful approximation tool. They may be used with a minimal number of basis functions for synthesizing the functions that are poorly approximated by other methods. Wavelet analysis allows you to investigate the properties of a signal in the time and frequency domains.

The wavelet transform may be used in systems with identifier [29]. The expediency of using it for the identification of nonlinear systems with unknown time-varying coefficients, which can be represented as a linear combination of basic wavelet functions, was shown in [30].

Moreover, to solve identification problems, various wavelet types are used (biorthogonal wavelets, wavelet frames, wavelet networks, spline wavelets) [31].

## 2.8. Criteria of Linear System Stability in the Sence of the Spectrum of Multi-Scale Wavelet Expansion Analysis

A multiscale wavelet expansion of an associative predictive model of a nonlinear time-varying object (1) for the selected detailing level $L$ looks as follows [32]:

$$
\begin{aligned}
x(t) &= \sum_{k=1}^{N} c_{L,k}^{x}(t)\varphi_{L,k}(t) + \sum_{l=1}^{L}\sum_{k=1}^{N} d_{l,k}^{x}(t)\psi_{l,k}(t), \\
y(t) &= \sum_{k=1}^{N} c_{L,k}^{y}(t)\varphi_{L,k}(t) + \sum_{l=1}^{L}\sum_{k=1}^{N} d_{l,k}^{y}(t)\psi_{l,k}(t),
\end{aligned}
\tag{9}
$$

where $L$ is the depth of the multi-scale expansion ($1 \le L \le L_{max}$, where $L_{max} = \log_2 N^*$, and $N^*$ is the power of the set of states of the system in the system dynamics knowledgebase); $\varphi_{L,k}(t)$ are scaling functions; $\psi_{l,k}(t)$ are the wavelet functions obtained from mother wavelets by means of tension/compression and shift:

$$
\psi_{l,k}(t) = 2^{l/2}\psi_{\mathrm{mother}}\left(2^{l}t - k\right).
\tag{10}
$$

Haar wavelets are chosen as mother wavelets; $l$ is the level of data detailing; $c_{L,k}$ are the scaling coefficients, $d_{l,k}$ are the detailing coefficients. The coefficients are calculated by use of the Mallat algorithm. The object equation is as follows:

$$
\begin{aligned}
\sum_{k=1}^{N} c_{Lk}^{y}(t)\varphi_{Lk}(t) + \sum_{l=1}^{L}\sum_{k=1}^{N} d_{lk}^{y}(t)\psi_{lk}(t) &= \sum_{k=1}^{N}\left(\sum_{i=1}^{m} a_i c_{Lk}^{y}(t-i)\varphi_{Lk}(t-i)\right) + \\
+ \sum_{l=1}^{L}\sum_{k=1}^{N}\left(\sum_{i=1}^{m} a_i d_{lk}^{y}(t-i)\psi_{lk}(t-i)\right) &+ \sum_{k=1}^{N}\left(\sum_{s=1}^{S}\sum_{j=1}^{r_s} b_{sj} c_{Lk}^{s}(t-j)\varphi_{Lk}(t-j)\right) + \\
+ \sum_{l=1}^{L}\sum_{k=1}^{N}\left(\sum_{s=1}^{S}\sum_{j=1}^{r_s} b_{sj} d_{lk}^{s}(t-j)\psi_{lk}(t-j)\right)&.
\end{aligned}
\tag{11}
$$

By considering the detailing and approximating parts of (7) separately, we have

$$
d_{lk}^{y}(t)\psi_{lk}(t) = \sum_{i=1}^{m} a_i d_{lk}^{y}(t-i)\psi_{lk}(t-i) + \sum_{s=1}^{S}\sum_{j=1}^{r_s} b_{sj} d_{lk}^{s}(t-j)\psi_{lk}(t-j).
\tag{12}
$$

The sufficient conditions of the object (1) stability for $\forall k = \overline{1, N}$ for the detailing and approximating coefficients respectively are as follows [33]:

- if $m > R$, $R = \max\limits_{s=\overline{1,S}} r_s$:

$$
\begin{aligned}
\left|a_m d_{lk}^{y}(t-m)\right| &< \left|d_{lk}^{y}(t)\right|, \\
\left|a_m c_{Lk}^{y}(t-m)\right| &< \left|c_{Lk}^{y}(t)\right|,
\end{aligned}
\tag{13}
$$

- if $m < R$, $R = \max\limits_{s=\overline{1,S}} r_s$, then:

$$
\begin{aligned}
\left|\sum_{s=1}^{S} b_{sR} d_{lk}^{s}(t-R)\right| &< \left|d_{lk}^{y}(t)\right|, \\
\left|\sum_{s=1}^{S} b_{sR} c_{Lk}^{s}(t-R)\right| &< \left|c_{Lk}^{y}(t)\right|,
\end{aligned}
\tag{14}
$$

- if $m = R \neq 1$, $R = \max\limits_{s=\overline{1,S}} r_s$, then the condition of the stability for the detailing coefficients:

$$\left| a_m d_{lk}^y(t-m) + \sum_{s=1}^{S} b_{sm} d_{lk}^s(t-m) \right| < \left| d_{lk}^y(t) \right|, \tag{15}$$

for the approximating coefficients:

$$\left| a_m c_{Lk}^y(t-m) + \sum_{s=1}^{S} b_{sm} c_{Lk}^s(t-m) \right| < \left| c_{Lk}^y(t) \right|, \tag{16}$$

- if $m = R = 1$, $R = \max\limits_{s=\overline{1,S}} r_s$, then the condition of the stability for the detailing coefficients:

$$\left| a_1 d_{lk}^y(t-1) + \sum_{s=1}^{S} b_{s1} d_{lk}^s(t-1) \right| < \left| d_{lk}^y(t) \right|, \tag{17}$$

for the approximating coefficients:

$$\left| a_1 c_{Lk}^y(t-1) + \sum_{s=1}^{S} b_{s1} c_{Lk}^s(t-1) \right| < \left| c_{Lk}^y(t) \right|. \tag{18}$$

## 3. Determining Static Stability Degree by Gramian Method

The Gramian method [16] provides an effective tool for analyzing the stability degree of power systems. It enables the investigation of system dynamics on the basis of a new mathematical technique for solving Lyapunov and Sylvester equations [16]. The method is based on the decomposition of the Gramian matrix, which is the solution of Lyapunov or Sylvester equations, into the spectrum of the matrices of these equations. To study the stability of differential-algebraic equations describing a power system, the system Gramian is calculated in real time using the asymptotic Frobenius norms [15].

From the methods used for solving the discrete Lyapunov equation [34], we have chosen the one offered in [35]. It applies the Fourier transform and $z$-transform to the discrete Lyapunov equation. The solution of the Lyapunov equation is an integral in the complex area of the product of resolvents of two matrices: the dynamics matrix and its transposed and adjoint one.

Therefore, we investigate the stability of the linear model described above. Let the linear stationary discrete time-invariant system be as follows:

$$x(k+1) = Ax(k) + Bu(k), \; x(0) = 0, \tag{19}$$

$$y(k) = Cx(k),$$

where

$$x(k) \in \mathbb{R}^n, u(k) \in \mathbb{R}^m, \; y(k) \in \mathbb{R}^m.$$

Suppose the matrices $A_{\lceil n \times n \rceil}$, $C_{\lceil m \times n \rceil}$, $B_{\lceil n \times m \rceil}$ are the real ones, where $m$, $n$ integer positive numbers, $m \leq n$. Suppose that the system (19) is stable, fully controllable and observable; all matrix $A$ eigenvalues are distinct ones.

The system characteristics in the frequency domain are defined by the transfer function

$$H(z) = \frac{M_{n-1}z^{n-1} + \cdots + M_1 z + M_0}{N(z)}, \; M_j = CA_j B. \tag{20}$$

The methods of selective modal analysis (SMA) and normal forms (NF) are the closest ones to our approach. The most generic approach is based on the modal analysis (eigenvalue decomposition, EVD) and the use of a linearized power system model in the system operating point (SEP).

(1) SMA and NF employ a linear model of an autonomous system, while for the Gramian method, the model of a system with input actions is used.

(2) The main difficulty of calculations in the NF method is a formation of the initial conditions [15]. In the Gramian method, initial conditions are formed as the values of electrical and/or mechanical moments; some known input functions may be specified instead.

(3) In the Gramian method, the calculating of stability loss risk is reduced to calculating the sums of the energy functionals of dominant modes, while in the NF method, nonlinear interaction indices, which depend on the initial conditions, play the similar role.

(4) Dominant modes in the Gramian method are determined by the participation factor of the energy functional of the mode in the total value of the square of $H_2$ norm of the power system's discrete transfer function [4,9]. The NF method uses nonlinear modal persistence indices for estimating the extent of dominance of the mode combinations for the third-order continuous approximating model.

(5) Modern electric power systems feature high dimension of the tasks being solved. The main method for constructing an approximating model for such systems is the interpolation based on the use of controllability and observability Gramians for linear and bilinear systems [8–10,34].

Suppose that transfer function (20) is strictly proper. Consider the following algebraic discrete Lyapunov (Stein) equation of the form [34]

$$\begin{aligned} AP^c A^* + BB^* &= P^c, \\ A^* P^o A + C^* C &= P^o. \end{aligned} \tag{21}$$

It is known that the matrix $A$ resolvent decomposition has the form:

$$[(Iz - A)]^{-1} = \sum_{j=0}^{n-1} z^j A_j N^{-1}(z), \tag{22}$$

where $N(z)$ is characteristic polynomial of $A$. The matrices $A_{j[n \times n]}$, called *Fadeev matrices* [36], can be defined by means of Fadeev–Leverie algorithms [37]. Suppose that $a_n = 1$, $A_{n-1} = I$.

Then the algorithm has the form

$$a_{n-k} = -\frac{1}{k} tr(AA_{n-k}), \ A_{n-k-1} = -a_{n-k}I + AA_{n-k}, \ k = 1, 2, \cdots, n. \tag{23}$$

The solutions of Equation (19) in the time domain can be defined in the following way [38]:

$$P^c = \sum_{k=0}^{\infty} A^k BB^* (A^*)^k, \ P^o = \sum_{k=0}^{\infty} (A^*)^k C^* CA^k. \tag{24}$$

The solution in frequency domain looks as follows:

$$P^c = \frac{1}{2\pi} \int_0^{2\pi} \left(e^{-i\theta} - A\right)^{-1} BB^* \left(e^{-i\theta} - A^*\right)^{-1} d\theta, \tag{25}$$

$$P^o = \frac{1}{2\pi} \int_0^{2\pi} \left(e^{-i\theta} - A^*\right)^{-1} C^* C \left(e^{-i\theta} - A\right)^{-1} d\theta. \tag{26}$$

The following variable change $e^{i\theta} = z$ is made in the integrals:

$$P^c = \frac{1}{2\pi i} \int_\gamma \left(z^{-1}I - A\right)^{-1} BB^* z^{-1} \left(zI - A^*\right)^{-1} dz, \tag{27}$$

$$P^o = \frac{1}{2\pi i} \int_\gamma \left(z^{-1}I - A^*\right)^{-1} C^* C z^{-1} (zI - A)^{-1} dz, \tag{28}$$

where $\gamma$ is a unit circle, moving counter-clockwise. Therefore, all eigenvalues are distinct ones, and we have the following resolvent decomposition:

$$
\begin{aligned}
(Iz - A)^{-1} &= \sum_{j=0}^{n-1} \sum_{k=1}^{n} \frac{z_k^j A_j}{\dot{N}(z_k)} \frac{1}{z - z_k}, \\
\left(Iz^{-1} - A\right)^{-1} &= \sum_{j=0}^{n-1} \sum_{k=1}^{n} \frac{z_k^{-j} A_j}{\dot{N}(z_k)} \frac{1}{z^{-1} - z_k}.
\end{aligned}
\tag{29}
$$

We introduce the following designations:

$$\left[ Res(zI - A)^{-1}, z = z_k \right] = \frac{z_k^j A_j}{\dot{N}(z_k)} = R_k, \tag{30}$$

$$BB^* = Q_d = \left[q_{d\rho,k}\right], \quad C^*C = W_d = \left[w_{d\rho,k}\right]. \tag{31}$$

After substituting this formula in Equations (27) and (28) we obtain:

$$P^c = \frac{1}{2\pi i} \int_\gamma \sum_{k=1}^{n} R_k \frac{1}{z - z_k} BB^* z^{-1} \left(z^{-1}I - A^*\right)^{-1} dz, \tag{32}$$

$$P^o = \frac{1}{2\pi i} \int_\gamma \sum_{k=1}^{n} z^{-1} \left(z^{-1}I - A^*\right)^{-1} C^* C R_k \frac{1}{z - z_k} dz. \tag{33}$$

Let us introduce the following designations for each sub-Gramian:

$$P_k^c = \frac{1}{2\pi i} \int_\gamma R_k \frac{1}{z - z_k} BB^* z^{-1} \left(z^{-1}I - A^*\right)^{-1} dz, \tag{34}$$

$$P_k^o = \frac{1}{2\pi i} \int_\gamma z^{-1} \left(z^{-1}I - A^*\right)^{-1} C^* C R_k \frac{1}{z - z_k} dz. \tag{35}$$

The integrands in Equations (34) and (35) are the analytic functions over the whole complex plane with the exclusion of particular points $= z_k$. By means of the Caushy residue theorem, we have the Gramians spectral semi-decomposition in following way:

$$P^c = \sum_{k=1}^{n} P_k^c, \quad P_k^c = R_k BB^* \left[z^{-1} \left(z^{-1}I - A^*\right)^{-1}\right]_{z - z_k}, \tag{36}$$

$$P^o = \sum_{k=1}^{n} P_k^o, \quad P_k^o = \left[z^{-1} \left(z^{-1}I - A^*\right)^{-1}\right]_{z - z_k} C^* C R_k. \tag{37}$$

We transform the matrix $\left[z^{-1} \left(z^{-1}I - A^*\right)^{-1}\right]_{z - z_k}$ by using matrix resolvent decomposition to vulgar fractions [39]:

$$
\begin{aligned}
\left[z^{-1} \left(z^{-1}I - A^*\right)^{-1}\right]_{z - z_k} &= \frac{1}{z_k} \lim_{z^{-1} \to z_k} \left(z^{-1} - z_k\right)\left(z^{-1}I - A^*\right)^{-1} = \\
&= \frac{1}{z_k} \left[ Res\left(z^{-1}I - A^*\right)^{-1}, z^{-1} = z_k \right] = \sum_{\rho=1}^{n} \sum_{j=0}^{n-1} \frac{z_\rho^{-j} A_j^*}{\dot{N}(z_\rho)} \frac{1}{1 - z_k z_\rho}.
\end{aligned}
\tag{38}
$$

By substituting the Equations (36) and (37) to the above formulae, we obtain the full decomposition of the Gramians in the form:

$$P^c = \sum_{k=1}^{n} \sum_{\rho=1}^{n} P_{k,\rho}^c, \; P_{k,\rho}^c = \sum_{\eta=0}^{n-1} \sum_{j=0}^{n-1} \frac{z_k^\eta z_\rho^{-j}}{\dot{N}(z_k)\dot{N}(z_\rho)} \frac{1}{1 - z_\rho z_k} A_j BB^* A_\eta^*, \tag{39}$$

$$P_{k,\rho}^c = \frac{1}{1-z_\rho z_k} R_k B B^* R_\rho, \tag{40}$$

$$P^o = \sum_{k=1}^{n} \sum_{\rho=1}^{n} P_{k,\rho}^o, \quad P_{k,\rho}^o = \sum_{\eta=0}^{n-1} \sum_{j=0}^{n-1} \frac{z_k^\eta z_\rho^{-j}}{\dot{N}(z_k)\dot{N}(z_\rho)} \frac{1}{1-z_\rho z_k} A_\eta^* C^* C A_j, \tag{41}$$

$$P_{k,\rho}^o = \frac{1}{1-z_\rho z_k} R_k C^* C R_\rho. \tag{42}$$

**Theorem 1.** *Consider LTI MIMO—the real discrete system with (A, B, C) presentation in the form where the matrices A, B, C are the real ones. Suppose the system is stable, fully controllable and observable; its transfer function is strictly proper, and the matrix A eigenvalues are distinct ones.*

Then the following matrix equalities hold:

$$P^c = \frac{1}{2\pi i} \int_\gamma \sum_{k=1}^{n} R_k \frac{1}{z-z_k} B B^* z^{-1} \left(z^{-1}I - A^*\right)^{-1} dz \quad \forall z : |z| < 1, \tag{43}$$

$$P^o = \frac{1}{2\pi i} \int_\gamma \sum_{k=1}^{n} z^{-1} \left(z^{-1}I - A^*\right)^{-1} C^* C R_k \frac{1}{z-z_k} dz \quad \forall z : |z| < 1, \tag{44}$$

$$P_k^c = \frac{1}{2\pi i} \int_\gamma R_k \frac{1}{z-z_k} B B^* z^{-1} \left(z^{-1}I - A^*\right)^{-1} dz \quad \forall z : |z| < 1, \tag{45}$$

$$P_k^o = \frac{1}{2\pi i} \int_\gamma z^{-1} \left(z^{-1}I - A^*\right)^{-1} C^* C R_k \frac{1}{z-z_k} dz \quad \forall z : |z| < 1, \tag{46}$$

where $R_k$ is the residue of the matrix $A$ resolvent in the point equal to the matrix eigenvalue. The Equations (43)–(46) define the semi-decomposition of the infinite controllability and observability Gramians on the eigenvalues set to belong of unit circle:

$$P^c = \sum_{k=1}^{n} \sum_{\rho=1}^{n} P_{k,\rho}^c, \quad P_{k,\rho}^c = \sum_{\eta=0}^{n-1} \sum_{j=0}^{n-1} \frac{z_k^\eta z_\rho^{-j}}{\dot{N}(z_k)\dot{N}(z_\rho)} \frac{1}{1-z_\rho z_k} A_j B B^* A_\eta^* \quad \forall z : |z| < 1, \tag{47}$$

$$P_{k,\rho}^c = \frac{1}{1-z_\rho z_k} R_k B B^* R_\rho \quad \forall z : |z| < 1, \tag{48}$$

$$P^o = \sum_{k=1}^{n} \sum_{\rho=1}^{n} P_{k,\rho}^o, \quad P_{k,\rho}^o = \sum_{\eta=0}^{n-1} \sum_{j=0}^{n-1} \frac{z_k^\eta z_\rho^{-j}}{\dot{N}(z_k)\dot{N}(z_\rho)} \frac{1}{1-z_\rho z_k} A_\eta^* C^* C A_j \quad \forall z : |z| < 1, \tag{49}$$

$$P_{k,\rho}^o = \frac{1}{1-z_\rho z_k} R_k C^* C R_\rho \quad \forall z : |z| < 1, \tag{50}$$

and the Equations (39)–(42) define full decomposition of the infinite Gramians on the eigenvalues set to belong to the interior of the unite circle on complex plane. Expressions for Gramians spectral decomposition one can simplify *Corollary* [38]. If all eigenvalues of the matrix $A$ are distinct, then the matrix can be transformed to diagonal form by means of the similarity transform:

$$\begin{aligned} x_d &= \mathrm{T}x, \quad x_d(k+1) = \Lambda x_d(k) + B_d u(k), \quad y_d(k) = C_d x_d(k), \\ \Lambda &= TAT^{-1}, \quad B_d = TB, \quad C_d = CT^{-1}, \end{aligned} \tag{51}$$

or

$$A = \begin{bmatrix} u_1 & u_2 \cdots & u_n \end{bmatrix} \begin{bmatrix} z_1 & 0 & 0 & 0 \\ 0 & z_2 & & 0 \\ 0 & & \ddots & \\ 0 & 0 & & z_n \end{bmatrix} \begin{bmatrix} v_1^* \\ v_2^* \\ \vdots \\ v_n^* \end{bmatrix} = T^{-1}\Lambda T, \ TV = VT = I, \tag{52}$$

where matrix $T^{-1}$ consists of the right eigenvectors $u_i$, and matrix $T$ consists of the left eigenvectors $V_i^*$ corresponding to the eigenvalue $s\, z_i$. The last equality is a condition for eigenvectors normalization.

The Gramians of diagonalized system are the solution of Lyapunov equations:

$$\Lambda P_d^c \Lambda^* + B_d B_d^* = P_d^c, \tag{53}$$

$$\Lambda^* P_d^o \Lambda + C_d^* C_d = P_d^o, \tag{54}$$

which are defined from the formulae:

$$P_{dk,\rho}^c = \frac{1}{1 - z_\rho z_k} R_k B_d B_d^* R_\rho \ \forall z : |z| < 1, \tag{55}$$

$$P_{dk,\rho}^o = \frac{1}{1 - z_\rho z_k} R_k C_d^* C_d R_\rho \ \forall z : |z| < 1. \tag{56}$$

The controllability Gramian $P_d^c$ is linked with the Gramian $P^c$ by equation

$$P^c = T^{-1} P_d^c \left(T^{-1}\right)^*. \tag{57}$$

The observability Gramian $P_d^c$ is linked with Gramian $P^c$ by similar equation

$$P^o = T^* P_d^o T. \tag{58}$$

We introduce the new designation $\mathbf{1}_{ij}$ for the matrix with all zeros except for the element "$ij$", which is equal to one (1):

$$\mathbf{1}_{ij} = \begin{bmatrix} 0 & 0 & \cdots & 0 & 0 \\ 0 & 0 & 0 & & 0 \\ \vdots & 0 & 1 & 0 & \vdots \\ 0 & & 0 & & 0 \\ 0 & 0 & \cdots & 0 & 0 \end{bmatrix}. \tag{59}$$

For diagonalized matrix $A$, the following expressions are valid:

$$(Iz - \Lambda)^{-1} = \sum_{k=1}^{n} R_k (z - z_k)^{-1} = \sum_{k=1}^{n} \mathbf{1}_{kk}(z - z_k)^{-1}, R_k = \mathbf{1}_{kk}. \tag{60}$$

Consider the spectral decomposition of the controllability and observability Gramians by pairwise combinational spectrum of the dynamics matrix. In this case, the Equations (55) and (56) have the form:

$$P_{d\rho,k}^c = \frac{1}{1 - z_\rho z_k} \mathbf{1}_{\rho\rho} Q_d \mathbf{1}_{kk} = \frac{1}{1 - z_\rho z_k} \mathbf{1}_{\rho,k} q_{d\rho,k}, \tag{61}$$

$$P_{d\rho,k}^o = \frac{1}{1 - z_\rho z_k} \mathbf{1}_{\rho\rho} W_d \mathbf{1}_{kk} = \frac{1}{1 - z_\rho z_k} \mathbf{1}_{\rho,k} w_{d\rho,k}. \tag{62}$$

Note that the premultiplication of the matrix $Q_d$ by the matrix $\mathbf{1}_{\rho\rho}$ and the post-multiplication by the matrix $\mathbf{1}_{kk}$ allows us to cut from the matrix its element located in the intersection of the column "$k$" and the row "$p$". For the diagonalized system, we have the following formulae:

$$P_d^c = \bigoplus \sum_{\rho,k} \mathbf{1}_{\rho,k} \frac{q_{dk,\rho}}{1 - z_\rho z_k}, \quad P_d^o = \bigoplus \sum_{\rho,k} \mathbf{1}_{\rho,k} \frac{w_{dk,\rho}}{1 - z_\rho z_k}. \tag{63}$$

These simple and compact expressions allow to compute sub-Gramians by computing their $n^2$ elements. They are simpler than the common Equations (47)–(50) for Gramian spectral expansion. We have got spectral separable Gramians decomposition in the form of a direct sum of $n^2$ sub-Gramians corresponding to the decomposition of controllability and observability Gramians by pairwise combinational eigenvalues of the dynamics matrix's spectrum.

As it is known [34], the necessary and sufficient condition for the energy stability of the system in terms of the square of the $H_2$ norm of the transfer function has the form:

$$\|\mathbf{G}(z)\|_2^2 < +\infty. \tag{64}$$

We define the stability loss risk functional as follows:

$$J(z_1, z_2, \ldots, z_n) = \|\mathbf{G}(z_1, z_2, \ldots, z_n)\|_2^2. \tag{65}$$

As the system approaches the stability threshold caused by the approaching of the characteristic equation roots to the imaginary axis, the risk functional approaches infinity. Let us define the acceptable risk of stability loss in the form

$$J(z_1, z_2, \ldots, z_n) = N_{perm}. \tag{66}$$

We will consider any system as *conditionally unstable* if all its roots are in the left half-plane, but the functional of the stability loss risk exceeds the established acceptable risk value. Accordingly, we will consider the system *conditionally stable* if

$$J(z_1, z_2, \ldots, z_n) < N_{perm}. \tag{67}$$

The square of the $H_2$ norm of the system transfer function can be calculated by solving the Lyapunov matrix algebraic equation by means of substituting the known matrices $A, B, C$ into it, while the spectrum of the matrix $A$ is not required to be calculated. On the other hand, the spectral expansions of the square of the $H_2$ norm of the system transfer function characterize the separability property of the stability loss risk functional: It is equal to the sum of terms, each one corresponding either to a separate eigenvalue of the dynamics matrix, or to their pairwise combination. The energy functional, which allows for only the weakly stable components of the quadratic forms $J_\delta$, makes it possible to determine the overall risk of stability loss as well as to estimate the energy stability margin in decibels:

$$M_{st} = 10 \cdot \lg \frac{N_{perm}}{J} \mathrm{dB} \approx 10 \cdot \lg \frac{N_{perm}}{J_\delta} \mathrm{dB}. \tag{68}$$

The mathematical model of the system is linear; however, the spectral decomposition of the square of H$_2$—norm of the power system discrete transfer function takes into account the nonlinear interaction of modes. The group interaction of modes is limited by taking into account only pairwise combinations of eigenvalues of the dynamics matrix.

The Gramians method can be used simultaneously for state monitoring and control of large-scale power systems, in particular, for static stability analysis; for developing stability estimator; for detecting dangerous free and forced oscillations; and for assessing the resonant interaction of dangerous oscillations [38,39].

## 4. Case Studies

*4.1. Automatic Remote Diagnostics of the Readiness for General Primary Frequency Control in a Power System*

Dynamic properties of power systems can be analyzed on the basis of the transient mode monitoring technique implemented in the special equipment for real-time information recording and transmitting. Modern primary frequency control systems contain digital models of generator excitation, turbine speed controllers, dynamic load models, and protection and automation models. In the event of an emergency situation resulting in a voltage drop in the network, all power plants carry out primary frequency control by changing the power through automatic turbine unit speed controllers, and the capacity of boilers, nuclear reactors, etc.

Primary frequency control should be carried out by power plants that have the prescribed primary control characteristics. The participation of a specific power unit in the primary frequency control is determined by a set of parameters that can change over the equipment life. If the characteristics of generating facilities do not meet the conventional standards, this can result in the stability loss of the entire power system in case of high frequency fluctuations. Therefore, it is very important for the normal operation of the power system to diagnose the current state of the facilities and their readiness for primary frequency control [21].

The only diagnostic technique available today is inspection test. However, for the entire testing period, it is required to disable the generation process that is too expensive. We propose a technique for remote diagnostics of the readiness of generating facilities for primary frequency control per aggregated responses to sudden frequency changes during normal operation. Check tests include checking the speed governors for each turbine; joint testing of the power unit; testing a section of a heat block with a common steam pipe. The main requirements to be met by generating equipment are as follows:

- The whole of main and auxiliary equipment, automation devices for power units, power plants, and their operating modes should allow within the prescribed load limits the amplitude of the primary control up to 20% of the rated power;
- When the power of the turbine unit changes in the range of ±10% of the nominal value, the power value is formed by the basic and auxiliary equipment, as well as the process automation equipment of the power unit/station. In this case, the speed governor must provide the specified transient time.

The regulatory control of operating mode parameters (the position of the turbine governor, the inlet pressure of the steam turbine, etc.) should ensure the proximity of the experimental transient response of the primary frequency control to the required one (which is verified during certification tests).

The results of certification tests (empirical transient characteristics and parameter estimates), as well as the type of turbine, frequency slope and deadband of turbine speed controllers, slopes and deadbands of frequency shifts of power controllers, altogether make the content of the knowledgebase of the automatic diagnostic system.

An aggregated dynamic model providing parameter estimates as well as static and dynamic properties of certain parameters and their relationships is created on the basis of the associative search technique (Figure 3).

Monitoring the changes in the parameters of associative models for turbine unit identification, which were developed using the raw data from a real-life power system, makes it possible to identify and predict negative trends in the operation of a specific unit and the performance of its control system deteriorating the overall performance of the primary frequency control system. Source: illustration of authors and collaborators.

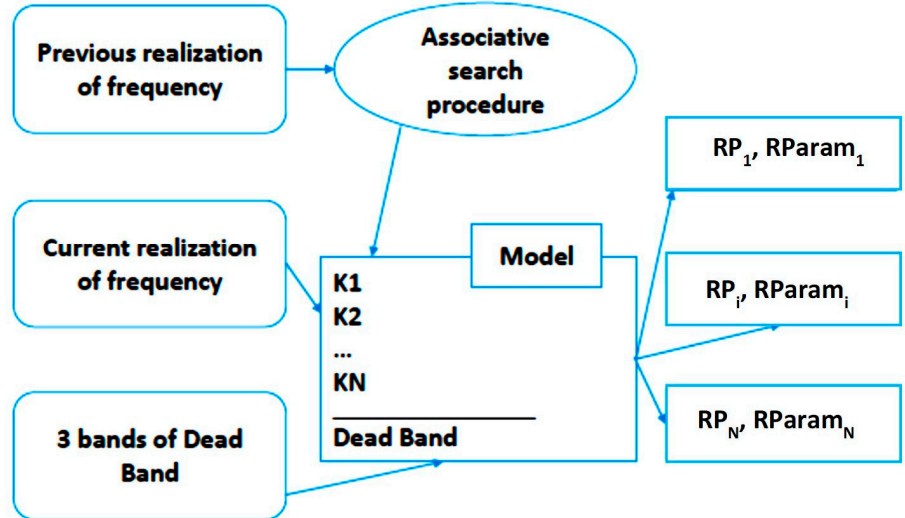

**Figure 3.** Frequency analysis by intelligent model. $RP_1, \ldots RP_i, \ldots RP_N$ power realizations values, K1, $\ldots$ KN linear model coefficients.

### 4.2. Full Power Forecast

The data of the overhead power line Kostroma Hydroelectric Power Plant (HPP)—Zagorka Pumped Storage Plant (PSPP) [21] were used for model development. (PSPP is a hydroelectric power plant that allows to mitigate daily electrical load changes). Transient monitoring data were collected using PMUs (Phasor Measurement Units) on 500 kV transmission lines. The sample was taken during 20 ms every 10 min.

We analyze and predict the full power. The load actually imposed on the consumer is described. This is the load imposed on electrical power elements (wires, cables, cabinets, transformers, power lines). Full AC power, which determines the currents and voltages, consists of an active component and inactive components of the power transmitted to the load (reactive, distortion and asymmetry) and can be expressed as follows:

$$S(t) = a_0 + a_1 S(t-1) + b_1 Ua(t-1) + b_2 Ia(t-1), \tag{69}$$

where $S(t)$ is the full output forecast; $S(t-1)$ is the current total power; $Ua(t-1)$ is the current value of the phase voltage, $Ia(t-1)$ is the current phase value of the current, $a_1$, $a_2$, $b_1$—are the model coefficients.

Figure 4 shows the accuracy advantage of the associative model over the traditional linear model. The conception of the authors and their collaborators was illustrated by Mathlab modeling. Process forecast was obtained with the help of conventional linear and associative models running for 43.7 s (2185 time steps, 20 ms per each step).

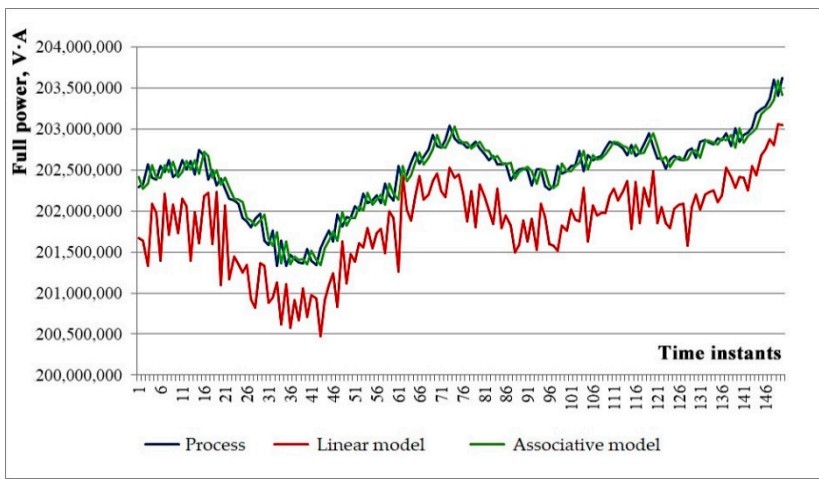

**Figure 4.** The forecast of the full power.

## 5. Conclusions

In this paper, we investigated the possibility of combining associative search identification technique with the Gramian method for predicting of the approaching of process dynamics to the stability threshold. A new technique for solving discrete Lyapunov equations based on matrix equations and semi-expansions of controllability Gramians was developed.

Features and novelty of the Gramian method are as follows:

(1) New methods of elementwise computation of the algebraic Lyapunov equation solution, based on separable spectral expansions of the controllability and observability Gramians;

(2) New energy criteria for assessing the risk of the loss of electric power systems stability, based on the identification and analysis of nonlinear effects caused by modes interaction;

(3) The method for identifying any potential swing centers and the forecast of the evolution of swing processes caused by the interaction of the modes in the linear model.

In the analysis of static and transient stability in power systems, the Gramian method occupies an intermediate position between the methods of selective modal analysis and normal forms. From the first one, it differs by the fact that it is actually a nonlinear modal analysis technique because the spectral expansions of Gramians in pairwise mode combinations include the products of second order infinitesimals. It also differs from the second one owing to the capability of (i) obtaining a direct assessment of stability loss risk based on the use of energy functionals and (ii) predicting the evolution of the stability loss process.

A linear discrete model is developed for predicting the approach of the state of a nonlinear system to the stability threshold. To build such a model, identification methods and algorithms based on knowledge formation and analysis are proposed and named associative search algorithms.

It is shown how the proposed method uses data mining for performing dynamic predictive remote diagnostics for the general primary frequency control in a power system. The application cases discussed in the paper demonstrate the high accuracy of the estimates obtained with help of associative search algorithm. The results of the theoretical investigation of identification models, algorithms, and methods developed by the authors were applied in the Scientific and Technical Center of the Russian Federal Grid Company.

Before applying the Gramian method, the approach of the system to the stability threshold can be predicted on the basis of multiresolution wavelet decompositions. The features and novelties of the proposed associative search algorithms are as follows:

(1) They allow to obtain linear models of nonlinear objects (at any given time step a new model is developed based on data mining);

(2) The version of algorithms for non-stationary objects is developed on the basis of wavelet analysis;

(3) New identification models feature high accuracy because they use the maximum of available information about object operation.

The difficulties of predictive model development and the analysis of their stability for electric power systems requires a systematic approach. Therefore, combining the methods proposed by the authors for constructing identification models with Gramians methods for stability studying at the stages of planning, monitoring, management, and optimization provides a synergistic effect and opens up a wider application outlook.

**Author Contributions:** Conceptualization, N.B. and I.Y.; methodology, N.B. and I.Y.; software, N.B. and I.Y.; validation, N.B. and I.Y.; formal analysis, N.B. and I.Y.; investigation, N.B. and I.Y.; data curation, N.B. and I.Y.; writing—original draft preparation, N.B. and I.Y.; writing—review and editing, N.B. and I.Y.; visualization, N.B. and I.Y.; supervision, N.B. and I.Y.; project administration, N.B. and I.Y.; funding acquisition, I.Y. All authors have read and agreed to the published version of the manuscript.

**Funding:** This research was funded by Russian Science Foundation, grant number 19-19-00673. The APC was funded by Russian Science Foundation.

**Conflicts of Interest:** The authors declare no conflict of interest.

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
