# Peer review of "Discrete Predictive Models for Stability Analysis of Power Supply Systems"

_mathematics, doi:10.3390/math8111943_

Round 1
Reviewer 1 Report
This paper aims to develop an approach to analyse the dynamics of the nonlinear time variant dynamic process. In general, this paper addresses a challenging topic. However, the major issue of this manuscript is the lack of original contributions. Detailed comments can be found below:
(1) Contribution and Introduction: the major issue of the paper is the difficulty in finding the original contributions. One of the reasons is the current Introduction is not sufficient (only 7 refs in the current Introduction), and more references need to be reviewed to clearly show the research gap. The authors should try to answer the following questions in Introduction: Where do we stand today? What seems to be the best methods/models? Have they been properly designed? An updated and complete literature review should be conducted to present clear knowledge gaps in the topic. Please also avoid using the form of [X-Y] to cite the references. In the end of Introduction, the contributions are suggested to be highlighted point by point in a straightforward way.
(2) Abstract: The Abstract is poorly written. The aim and contributions cannot be found after reading the abstract. Numerical results are expected in the Abstract as well as in Conclusion section.
(3) Structure: The manuscript was not well structured and was not prepared in a professional manner. The first impression of the paper is like a textbook, and not compact enough. A large number of paragraphs only contain 1-2 sentences. It is also quite confusing to see loads of equations in Section 4 after the section of case study (Section 3).
(4) Language: A native English speaker is recommended to help remove typos and grammatical errors and improve the readability of the manuscript.
Author Response
Many thanks for your comments!
See file in attachment, please.

Reviewer 2 Report
The manuscript presents the identification techniques and algorithms based on knowledge processing forming and analysis. Different case studies have been discussed to demonstrate the high accuracy of the estimates obtained with help of associative search algorithm. The topic of the paper fits the aims and scope of the "mathematics" and the contribution is useful for researchers and readers in the field. However, there are some major rooms that should be addressed by the authors before the paper could be considered for publication:
Different sections of the paper are written very brief and suffer from the lack of detailed information:
- The abstract section should be extended to cover all achievements of the manuscript.
- In the introduction section, an analytical gap analysis should be done to justify the necessity of doing this piece of research.
- The results analysis should be extended with more details; especially about the limitations and assumptions of the method as well as its relative merits to other publicly available proposals.
Author Response
Many thanks for your comments.
See file in attachment, please.

Reviewer 3 Report
Overall, the paper is well written and analyzes an important aspect of contemporary reality, that of power supply system in industry.
However, analyzing this paper in detail, I have two important remarks:
- Each scientific paper must contain the following sections - "Introduction", “Literature review” (or “Background”), “Research methodology”, “Calculation / analysis”, “Results & Discussion ” and “Conclusions”. From. this point of view, the author only partially respects this structure presented above. Therefore, I recommend a remodeling / rewriting of the paper, taking into account the sections described above.
- Every figure / table must present the source. If the table or the table is the result of the author research, than it should be mention like this : “Source: Author own conception, based on XYZ software”.
- A scientific article must not end with bullets and numbers (“3) new models are significantly more accurate than known identification models (see Fig.4), because associative search algorithms use all a priori information about an object – in the form of knowledge: regularities, patterns.”). Otherwise, it looks like a scheme or a draft version of the article.
- The author must present the limitations of the research study; also, in order to improve the quality of the research (and to give weight to the article), the author must emphasize more clearly the practical implications of the conducted research.
- The “References” section must be improved, adding 10 – 15 titles at least.
- The author does not present the novelty degree of the conducted research.
Author Response

(The authors gave the same response as above.)

Reviewer 4 Report
The paper approaches an interesting subject! Congratulations!
All the equations should be numbered.
The quantities should occur on the Figure 4 axes.
Apart from these remarks, I consider that the paper can be published in the present form.
Author Response

(The authors gave the same response as above.)

Round 2
Reviewer 1 Report
The authors have addressed the comments appropriately. The current version is recommended for acceptance.